The clinical value of the Duke Anesthesia Resistance Scale in predicting postoperative delirium after hip fracture surgery: a retrospective study

Wang Yaya
Jiang Yan’an
Fu Huajun
Zhao Yikang
Xu Zhao xu_zhao996@126.com
Department of Anesthesiology, Shaanxi Provincial People’s Hospital, Third Affiliated Hospital of Xi’an Jiaotong University , Xi’an , China
Qin Jiangjiang
Electronic publication date: 2023 Dec 4
Publication date: 2023
Volume: 11
Electronic Location ID: e16535
Received 2023 Aug 21; Accepted 2023 Nov 7
Copyright: © 2023 Wang et al.
Copyright year: 2023
Copyright holder: Wang et al.
License: This is an open access article distributed under the terms of the Creative Commons Attribution License, which permits unrestricted use, distribution, reproduction and adaptation in any medium and for any purpose provided that it is properly attributed. For attribution, the original author(s), title, publication source (PeerJ) and either DOI or URL of the article must be cited.
License URL: https://creativecommons.org/licenses/by/4.0/

Keywords: Duke Anesthesia Resistance Scale, Hip fractures, Delirium, Predictive value

Funding: Shaanxi Provincial People’s Hospital Science and Technology Development Incubation Fund 2022YJY-34 This study was supported by the Shaanxi Provincial People’s Hospital Science and Technology Development Incubation Fund (2022YJY-34). The funders had no role in study design, data collection and analysis, decision to publish, or preparation of the manuscript.

==============================
Aim

This study aims to investigate the clinical value of the Duke Anesthesia Resistance Scale (DARS) in predicting postoperative delirium (POD) after hip fracture surgery.

Methods

A retrospective study was conducted. Clinical data were collected from the patients who had hip fracture and underwent elective total hip arthroplasty in Shaanxi Provincial People’s Hospital, Third Affiliated Hospital of Xi’an Jiaotong University between January 2022 and June 2023. The Consciousness Fuzzy Assessment Scale was used to evaluate the occurrence of POD on postoperative day 3 (POD 3). The enrolled patients were divided into the POD group (n = 26) and the non-POD group (n = 125). Baseline characteristics, surgical data, postoperative information, and laboratory test results were collected. DARS scores were calculated using the minimum alveolar concentration, end-tidal concentration average (ETAC), and bispectral index (BIS). Multivariate logistic regression analysis was conducted to recognize the independent risk factors for POD after hip fracture surgery. Receiver operating characteristic (ROC) curve was plotted to evaluate the value of DARS in POD prediction.

Results

The average age of POD group was significantly higher, comparing to non-POD group (P < 0.05). DARS scores were statistically lower in the POD group compared to non-POD group (P < 0.05). Multivariate logistic regression analysis found that age and DARS scores were factors impacting post-operative delirium occurrence after hip fracture surgery (P < 0.05). ROC showed that the area under the curve for DARS in predicting POD after hip fracture surgery was 0.929 (95% CI [0.861–0.997]). The optimal cutoff value was 30. The sensitivity was 95.45%, while the specificity was 84.09%.

Conclusion

DARS score demonstrates good predictive value in hip fracture patients and is feasible in clinical practice, making it suitable for clinical application and promotion.

Introduction

Hip fractures are more common in the elderly population, primarily due to osteoporosis, leading to falls and subsequent fractures. According to statistics (Veronese & Maggi, 2018), approximately 1.5 million people worldwide are hospitalized each year for hip fractures. Currently, surgical intervention is the main method for clinical treatment of hip joints, effectively aiding patients in hip joint functionality recovery and improving their quality of life (Chen et al., 2022). Postoperative delirium, characterized by cognitive impairment and other related symptoms, is a common complication following hip fractures, with an incidence rate ranging from 13% to 70% (Wu et al., 2021). It significantly impacts the postoperative recovery of patients. Studies have shown that delirium is a considerable impacting factor contributing to inferior prognosis in hip fracture patients. In severe cases, it can even increase the risk of joint dislocation, prolong hospitalization, and impose additional medical burdens on patients (Zhao et al., 2023). Therefore, early prediction of postoperative delirium after hip fractures has significant importance in improving patient outcomes.

Previous research has indicated a close correlation between anesthesia depth and the occurrence of postoperative delirium (Wang et al., 2021). Thus, monitoring the Bispectral Index (BIS), adjusting the appropriate anesthesia depth accordingly, has shown positive significance in reducing the incidence rate of postoperative delirium. However, the application of BIS alone in predicting postoperative delirium has certain limitations due to factors such as muscle relaxants, ephedrine, and adrenaline (Liu et al., 2022). Cooter Wright et al. (2022) constructed the Duke Anesthesia Resistance Scale (DARS) based on the minimum effective end-expiratory alveolar concentration (aaMAC) and BIS.. DARS score is a method of brain anesthetic resistance measured by researchers in the Department of Anesthesiology of Duke University based on EEG, which has good value in reflecting postoperative delirium. The score mainly includes age-adjusted minimum effective end-expiratory alveolar concentration (aaMAC) and BIS. Studies have shown that (Cooter Wright et al., 2022), there is a nonlinear relationship between DARS and postoperative POD risk. The lower the DARS score, the higher the risk of delirium. However, its predictive value in postoperative delirium among Chinese patients with hip fractures remained unclear. This current study aims to investigate the application value of DARS in predicting delirium in postoperative patients with hip fractures, to provide reference for reducing the incidence rate of postoperative delirium.

Materials and Methods

Study population

A retrospective study was conducted to collect clinical data of patients who underwent elective total hip arthroplasty for hip fractures in Shaanxi Provincial People’s Hospital from January 2022 to June 2023. The inclusion criteria were as follows: (1) confirmed diagnosis of hip fracture; (2) age over 18 years; (3) scheduled for elective total hip arthroplasty; (4) American Society of Anesthesiologists (ASA) grade I-III; (4) medical records were completed.

The exclusion criteria was to exclude: (1) patients had significant organ dysfunction; (2) preoperative delirium; (3) comorbid psychiatric disorders; (4) history of coronary heart disease and ischemic cerebrovascular disease; (5) incomplete clinical data. In total, 151 patients were enrolled, including 84 males and 67 females. The range of age was 69–81 years old (mean, 73.89 ± 4.56 years). All samples obtained in this study were approved by the ethics committee of the Shaanxi Provincial People’s Hospital, Third Affiliated Hospital of Xi’an Jiaotong University and abided by the ethical guidelines of the Declaration of Helsinki, and ethics committee agreed to waive informed consent.

Based on whether postoperative delirium (POD) occurred within 3 days, patients were further divided into the POD group (n = 26) and the non-POD group (n = 125). POD definition : refers to the symptoms of delirium after surgery, characterized by changes in state of consciousness, cognitive dysfunction and abnormal behavior, which may be manifested as confusion, memory impairment, decreased orientation, inattention, hallucinations, delusions and other symptoms.The diagnostic criteria for POD were as follows: POD was evaluated by trained researchers with the Chinese version of the Confusion Assessment Method (CAM) twice daily during within three postoperative days. The occurrence of POD was diagnosed if could be observed in any of the follow-up. Diagnosing criteria of POD were considered as: (1) acute fluctuating course, (2) attention impairment, (3) disorganized thinking, (4) altered consciousness level. The POD diagnosis required presenting both criteria (1) and (2), yet either criteria (3) or (4) (Goldberg et al., 2020).

Clinical procedure

Collection of clinical data

Clinical data of the patients were collected via electronic medical records, including (1) baseline characteristics (gender, age, body mass index (BMI), preoperative comorbidities, Mini-Mental State Examination (MMSE) score); (2) surgical details (ASA grade, operation time, time of anesthesia, blood loss intraoperatively); (3) postoperative data (hospital stay length, medication, new incidence of cardiovascular and cerebrovascular events within 28 days postoperatively, incidence rate of postoperative infections); (4) laboratory test results (preoperative albumin level, hemoglobin level, neutrophil count).

Anesthesia

All patients underwent relevant preoperative examinations, and upon admission, continuous monitoring of electrocardiogram, blood pressure, pulse oximetry, and urine output was conducted. Intravenous access was secured, administering 300–500 mL compound sodium lactate. Oxygen was administered via a face mask.

Anesthesia induction was performed using intravenous drugs, including rocuronium bromide (0.15 mg/kg), sufentanil (0.04–0.06 μg/kg), etomidate (0.2 mg/kg), followed by successful tracheal intubation.

Anesthesia maintenance was achieved by a combination of intravenous infusion and inhalation of sevoflurane at a concentration of 2–3%. The sevoflurane concentration was adjusted according to the patient’s condition. Intermittent administration of rocuronium bromide (0.1 mg·kg−1·min−1) was used to maintain neuromuscular blockade. Muscle relaxants were discontinued 30 min before the completion of surgery, and tracheal extubation was performed when extubation criteria were met. Patient-controlled intravenous analgesia (PCIA) was adopted for postoperative pain control, and further, a Numeric Rating Scale (NRS) score of <4 was used as the target for pain management. The PCIA formula included tramadol 100 mg, dexmedetomidine 0.3 μg/kg, tropisetron 8 mg, and isotonic saline solution (diluted to a total volume of 100 ml).

DARS score calculation method

The calculation method for the DARS score is described by the formula: DARS= (12.5−aaMAC)BIS. (1) The calculation method for the minimum alveolar concentration (aaMAC) could be found in reference (Cooter Wright et al., 2022). (2) The BIS values were obtained by continuously recording the BIS values from a multifunctional monitor produced by PHILIPS, with a signal quality index greater than 80%, calculating average of these values (Georgevici et al., 2021).

Statistical analyses

The data collected were analyzed by SPSS version 23.0 (SPS Inc., Chicago, IL, USA). Continuous variables that were normally distributed were demonstrated as mean ± standard deviation, t-tests were used for comparisons. Categorical variables were demonstrated as “number of cases (%)”, and chi-square tests were used for comparisons between groups. Variables that had a significance level of P < 0.05 in the univariate analysis were included in a multivariate logistic regression analysis model to analyze independent risk factors for POD. Receiver operating characteristic (ROC) curve was performed to evaluate the predictive value of DARS in predicting POD after hip fracture surgery. Statistical significance was calculated at a level of P < 0.05.

Results

Comparison of baseline characteristics between the two groups

No significant differences were detected in BMI, gender, as well as preoperative comorbidities between POD group and non-POD group (P > 0.05). However, the POD group had significantly higher average age compared to non-POD group (P < 0.05), as shown in Table 1.

Table 1 Baseline clinical characteristics comparison between groups.

	POD group
(n = 26)	Non-POD group
(n = 125)	t/2	P	
Age (years, x¯±s)	78.56 ± 7.31	63.31 ± 2.65	18.384	<0.001	
Gender (n, %)	Male	14 (53.85)	69 (55.20)	0.016	0.890	
	Women	12 (46.15)	56 (44.80)			
BMI (kg/m2, x¯±s)	24.79 ± 1.82	25.12 ± 1.94	0.799	0.426	
Preoperative comorbidities (n, %)	Hypertension	14 (53.85)	72 (57.60)	0.399	0.941	
	Diabetes	8 (30.77)	39 (31.20)			
	Hyperlipidemia	3 (11.54)	11 (8.80)			
	Others	1 (3.85)	3 (2.40)			
Preoperative
MMSE score ( x¯±s)	26.45 ± 1.26	26.23 ± 1.38	0.750	0.454	

Comparison of surgical data between the two groups

No significant differences were detected in ASA grade, anesthesia time, surgical time and intraoperative blood loss between POD group and non-POD group (P > 0.05). However, POD group had significantly lower DARS scores compared to the non-POD group (P < 0.05), as illustrated in Table 2.

Table 2 Surgical characteristics comparison between groups.

	POD group
(n = 26)	Non-POD group
(n = 125)	t/2	P	
ASA grade (n, %)	I	8 (30.77)	39 (31.20)	0.017	0.992	
	II	14 (53.85)	68 (54.40)			
	III	4 (15.38)	18 (14.40)			
Surgical time (min, x¯±s)	85.26 ± 16.25	84.41 ± 16.78	0.236	0.814	
Anesthesia time (min, x¯±s)	107.45 ± 30.56	105.21 ± 32.33	0.324	0.746	
Intraoperative
blood loss (ml, x¯±s)	274 ± 35	270 ± 38	0.495	0.622	
DARS score ( x¯±s)	23.32 ± 2.41	35.18 ± 5.69	10.413	<0.001	

Comparison of postoperative data between the two groups

No significant differences were detected in length of hospital stay, postoperative medication, occurrence of new cardiovascular and cerebrovascular events, and postoperative infection events between POD group and non-POD group (P > 0.05), as demonstrated in Table 3.

Table 3 Postoperative characteristics comparison between groups.

	POD group
(n = 26)	Non-POD group
(n = 125)	t/2	P	
Length of hospital stay (d, x¯±s)	10.89 ± 1.65	9.82 ± 2.11	0.017	0.992	
Postoperative medication (n, %)	Antihistamines	5 (19.23)	27 (21.60)	0.116	0.944	
Opioids	13 (50.00)	63 (50.40)			
Anticholinergic	8 (30.77)	35 (28.00)			
New cardio-and cerebrovascular events (n, %)	Cognition impairment	4 (15.38)	18 (14.40)	0.017	0.897	
Stroke	1 (3.85)	4 (3.20)	0.028	0.867	
	New onset atrial fibrillation	0	1 (0.80)	0.209	0.647	
Postoperative infection (n, %)	Pulmonary	1 (3.85)	2 (1.60)	0.558	0.455	
Wound	1 (3.85)	3 (2.40)	0.175	0.676	
Venous thrombosis (n, %)	1 (3.85)	3 (2.40)	0.175	0.676	

Comparison of laboratory test results between the two groups

No significant differences were found in preoperative albumin, hemoglobin, and neutrophil count between POD group and non-POD group (P > 0.05), as indicated in Table 4.

Table 4 Lab test results comparison between groups ( x¯ ± s).

	POD group
(n = 26)	Non-POD group
(n = 125)	t/2	P	
Albumin (g/L)	33.11 ± 7.34	35.69 ± 6.82	1.732	0.085	
Hemoglobin (g/L)	110.54 ± 2.37	111.27 ± 2.89	1.205	0.230	
Neutropil count (×109/L)	6.58 ± 2.13	6.41 ± 2.35	0.341	0.734	

Multivariable logistic regression analysis of POD impacting factors

Using the occurrence of postoperative POD as the dependent variable (1 = Yes, 0 = No), and age (continuous variable) and DARS scores (continuous variable) as independent variables, a multivariable logistic regression analysis was performed. The results revealed that both age and DARS scores were independent factors impacting POD occurrence postoperatively (P < 0.05), as demonstrated in Table 5.

Table 5 Multivariable logistic regression analysis of factors influencing postoperative POD events.

Risk factors	β value	SE value	Ward2	OR	95% CI	p	
Age	0.246	0.072	11.681	1.279	1.111~1.473	<0.001	
DARS score	−0.114	0.064	3.189	0.892	0.787~1.011	<0.001	

ROC curve analysis of DARS scores in predicting POD after hip fracture surgery

The ROC analysis showed that the area under the curve (AUC) for using DARS scores to predict postoperative POD in elderly patients, was 0.929 (95% CI [0.861–0.997]). The optimal cutoff value was determined to be 30, with a sensitivity of 95.45% and specificity of 84.09%, as shown in Fig. 1.

Figure 1 ROC curve for predicting POD after hip fracture surgery using DARS scores.

Discussion

POD, a reversible cognitive impairment, is characterized as acute decline in attention and cognitive function, considering as one of the adverse events that hinder patients’ postoperative recovery. The incidence of hip fractures is influenced by various factors, including age, gender, osteoporosis, lifestyle, and accidental injuries. With the accelerating pace of population aging, hip fractures are more common in the elderly, especially women over 50 years old. According to the National Osteoporosis Foundation (Weaver et al., 2016), about 50% of women will have fractures after the age of 65, including hip fractures. In this study, 26 out of 151 patients who underwent surgical treatment for hip fractures developed POD, with an incidence rate of 20.80%, consistent with previous literature reports (Bhushan et al., 2022). Currently, there are multiple theories regarding the causes of postoperative POD, including the psychosocial stress theory, inflammatory factor theory, and neurotransmitter theory (Li et al., 2019; Yang et al., 2022). However, the specific pathophysiological mechanisms are not yet clear. They might be closely related to decreased cerebral oxidative metabolism, particularly in the frontal lobe, and disturbances in central cholinergic deficiency and neurotransmitter regulation (Han et al., 2020). Studies have shown that the mortality rate within 6 months after hip fracture POD is approximately three times higher than that of patients without POD (Tao et al., 2019), indicating the importance of predicting postoperative POD in improving patient survival quality.

This study analyzed the factors influencing postoperative POD occurrence using univariate analysis and multiple logistic regression analysis. The results showed that age and DARS scores were impacting factors of POD occurrence (P < 0.001). Studies have shown, compared to patients under 60 years old, older patients have more pronounced decline in brain function, decreased organ function and physical adaptation ability, decreased reliability of physical regulatory systems, as well as slower metabolism and clearance of anesthetic drugs, leading to prolonged drug effects (Georgevici et al., 2021). Additionally, their sensitivity to stressors is further increased, resulting in enhanced stress responses and abnormal excitatory conduction, with increased risk of neurotransmitter disorders, thus increasing the incidence rate of POD. Literature has shown that for every 1-year increase in age, the incidence rate of POD increases by 2%, highlighting the importance of advanced age as a significant impacting factor of POD occurrence (Wang et al., 2018).

Numerous publications have demonstrated that anesthetic depth was closely related to occurrence of postoperative POD (Pérez-Otal et al., 2022; Li & Zhang, 2020). Excessive anesthesia may increase the risk of postoperative delirium, as patients may experience confusion and disorientation upon awakening after surgery. On the other hand, inadequate anesthesia depth, or insufficient anesthesia depth, may lead to conscious awakening, with patients consciously perceiving and remembering events during or after surgery. This situation may increase patient anxiety and fear related to the surgical experience, thereby increasing the risk of postoperative delirium. BIS is a monitoring index used to assess anesthesia depth. It quantifies anesthesia depth on a scale of 0 to 100 by analyzing the spectral characteristics of the electroencephalogram. Generally, a lower BIS value indicates a deeper anesthesia depth and a shallower level of consciousness, while a higher BIS value indicates a shallower anesthesia depth and a closer approximation to an awake state. The BIS index is commonly used as an auxiliary tool to guide the use of anesthetic drugs and manage anesthesia depth (Evered et al., 2021).

A study on the correlation between BIS value and postoperative delirium found that the lower the BIS value, the higher the incidence of postoperative delirium., possibly due to deeper anesthesia depth, which makes patients more prone to awakening, perception, or nociceptive stimulation, leading to anxiety, discomfort, and increased risk of postoperative delirium (Evered et al., 2021). This study confirmed the application value of BIS monitoring in predicting the occurrence of POD. However, it was found in clinical practice that the use of BIS alone for predicting postoperative POD has limitations, as the BIS index only serves as an indicator of anesthesia depth and cannot fully represent the patient’s level of consciousness. Anesthesia depth is also influenced by other factors, such as surgical stimulation, pain, and individual differences, which may result in insufficient predictive efficacy of the BIS index for POD (Chew et al., 2022).

The DARS score includes aaMAC in addition to the BIS value, while aaMAC is calculated based on two indicators, Age-corrected MAC and ETAC, both of which reflect the effective indicators of inhaled anesthetic depth (Whitlock et al., 2011; Cooter et al., 2020). The anesthesia depth monitor monitor monitors Age-corrected MAC and ETAC primarily by connecting a device called an end-tidal carbon dioxide probe at the end of the ventilator. The probe is located in the airway and can directly measure the concentration of carbon dioxide in the expiratory airflow, which can provide accurate concentration data of end-tidal carbon dioxide and minimum effective end-tidal alveolar concentration.MAC reflects the minimum alveolar concentration of inhaled anesthetic at an atmospheric pressure that prevents 50% of patients from responding to a noxious stimulus. Age-corrected MAC is used for evaluation due to the significant influence of age on MAC. Some studies (Hight et al., 2022) have shown that with the increase of age, the alveolar ventilation will decrease in general, and the anatomical and physiological characteristics of the respiratory tract will also change, which may affect the distribution and excretion rate of drugs in the alveoli. The sensitivity of the elderly to anesthetic drugs may increase, because the function of metabolism and drug clearance may decrease, which may lead to the need for lower drug concentrations to achieve the same anesthetic effect.The combined assessment of Age-corrected MAC, ETAC, and BIS may improve the accuracy of monitoring anesthesia depth and contribute to the improvement of anesthesia management. study showed (Cooter Wright et al., 2022) that there is a nonlinear relationship between DARS and postoperative POD risk. The lower the DARS score, the higher the risk of delirium. When the DARS threshold was 28.755, the Youden index of the association between low DARS and delirium reached the maximum, with 95% confidence intervals of 26.18 and 29.80. In the multivariate model, low DARS was associated with an increased risk of delirium (OR = 4.30, 95% CI [1.89–10.01], P = 0.001). In this study, the ROC curve is mainly drawn by GraphPad Prism 8 (GraphPad Software, La Jolla, CA, USA). The ROC curve is a commonly used method to evaluate the predictive value of the binary classification model. It represents the performance of the model by drawing the relationship curve between sensitivity and specificity. Ideally, the ROC curve should be as close to the upper left corner as possible; the area under the curve (AUC) is usually used to evaluate the predictive value of the model. The value range of AUC is between 0.5 and 1, and the closer the value is to 1, the better the prediction ability of the model is.This study evaluated the value of DARS scores in predicting POD occurrence using ROC curve analysis. The results showed that AUC for DARS in predicting postoperative POD in elderly THA was 0.929 (95% CI [0.861–0.997]). At this point, the sensitivity was 95.45% and the specificity was 84.09%, indicating the value of DARS scores serving as an auxiliary tool for predicting postoperative POD.

This study has some limitations. Most importantly, age remained a confounding factor that was not fully controlled for. Although age was included in the multivariate regression analysis, the broad age range of patients may have obscured its effects. The advanced age of many hip fracture patients is an inherent risk factor for postoperative delirium that cannot be eliminated. Our study population had a mean age of 73.89 years, with ages ranging from 69 to 81 years old. This wide distribution means that delirium risk from age-related factors likely varied substantially among the subjects. Controlling for age by analyzing subgroups of different age ranges could have helped isolate its confounding effects. However, the sample size of our study was too small to allow effective stratification by age group. In future studies, recruiting larger samples with a narrower age range would provide more definitive results on the relationship between DARS scores and postoperative delirium when accounting for differences in age-related delirium risk.

Conclusions

DARS score demonstrates good predictive value in hip fracture patients and is feasible in clinical practice, making it suitable for clinical application and promotion. However, this study is still a single-center study, while study subjects selection may have bias. Subgroup analysis of postoperative POD prediction using different anesthesia strategies and anesthesia drug doses was not conducted, which needs to be further improved and supplemented in the future study. Clinically, according to the changes of DARD score, targeted POD prevention countermeasures have been formulated, so as to effectively prevent the occurrence of POD after operation.

Supplemental Information

Supplemental Information 1 Raw Data.

Click here for additional data file.

Additional Information and Declarations

Competing Interests

Author Contributions

Human Ethics

Data Availability

The authors declare that they have no competing interests.

Yaya Wang conceived and designed the experiments, performed the experiments, prepared figures and/or tables, authored or reviewed drafts of the article, and approved the final draft.

Yan’an Jiang performed the experiments, prepared figures and/or tables, authored or reviewed drafts of the article, and approved the final draft.

Huajun Fu conceived and designed the experiments, analyzed the data, authored or reviewed drafts of the article, and approved the final draft.

Yikang Zhao analyzed the data, prepared figures and/or tables, and approved the final draft.

Zhao Xu conceived and designed the experiments, performed the experiments, analyzed the data, authored or reviewed drafts of the article, and approved the final draft.

The following information was supplied relating to ethical approvals (i.e., approving body and any reference numbers):

All samples obtained in this study were approved by the ethics committee of the Shaanxi Provincial People’s Hospital, Third Affiliated Hospital of Xi’an Jiaotong University and abided by the ethical guidelines of the Declaration of Helsinki.

The following information was supplied regarding data availability:

The raw data is available in the Supplemental File.

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
