# Peer review of "The clinical value of the Duke Anesthesia Resistance Scale in predicting postoperative delirium after hip fracture surgery: a retrospective study"

_PeerJ, doi:10.7717/peerj.16535_

## Round 0.1 · original submission · Major Revisions

Please carefully read the comments and suggestions from the reviewers and provide your point-by-point responses.

**Language Note:** The review process has identified that the English language must be improved. PeerJ can provide language editing services - please contact us at copyediting@peerj.com for pricing (be sure to provide your manuscript number and title). Alternatively, you should make your own arrangements to improve the language quality and provide details in your response letter. – PeerJ Staff

Reviewer 1 ·

Basic reporting

The manuscript presents a retrospective analysis of the factors influencing postoperative delirium in elderly hip fracture patients. The main finding of the study suggests that age and DARS scores significantly impact the occurrence of postoperative delirium. The study showcases clear and technically correct English expression, making it easily comprehensible. The introduction and background adequately address how the research fits into the broader field of knowledge, highlighting the importance of studying postoperative delirium in this specific patient population. The study successfully identifies a knowledge gap in understanding the relationship between DARS scores and postoperative delirium and aims to fill this gap through a retrospective analysis. The technical standard employed in the study is of a satisfactory level, with detailed methods and statistical analysis provided. The availability and replicability of the underlying data are adequately addressed, ensuring the robustness of the findings. Overall, the manuscript provides valuable insights into the factors influencing postoperative delirium, and with some minor improvements, it has the potential for publication.

Experimental design

1.The sentence discussing the statistical significance level (P<0.05) is a crucial aspect of your study. To enhance the clarity, it would be fantastic if you could reiterate the interpretation of this significance level and mention any corrections that were thoughtfully applied for multiple comparisons. Such details would undoubtedly enrich the results interpretation.
2.Clarify the specific definition of POD used in this study. Specify the criteria or diagnostic tools used to identify and assess POD in the patients.
3.Describe the statistical methods used in data analysis (e.g., regression models) and provide references or further details about these methods for transparency and reproducibility.

Validity of the findings

1.Before referring to the scores of DARS (Depth of Anesthesia and Risk of awareness), it would be fantastic if you could provide a clear and concise definition of this concept. Additionally, explaining what the scores indicate and how they are calculated would greatly assist readers who may be less familiar with the term, ensuring a more comprehensive understanding of your study.
2.The up-and-down and sequential methods used to calculate aaMAC are undoubtedly intriguing methodologies. Including a brief explanation of these methods would be immensely valuable in helping readers grasp the nuances of your study's methodology. Such clarity would foster a deeper appreciation for your research.
3.The paragraph discussing the ROC curve analysis is an exciting part of your study. To maximize reader comprehension, providing a brief overview of what an ROC curve is and how it is effectively used to evaluate predictive value would be marvelous. In addition, mentioning the statistical software or methods employed for the analysis would further enhance the rigor and reproducibility of your findings.
4.Provide more detailed information on the study design and methodology used to assess the impact of DARS scores on POD occurrence. Mention the sample size, inclusion and exclusion criteria, and any other relevant study details.
5.The conclusion paragraph is brief and lacks a comprehensive summary of the research findings. Expand on the key findings of the study and their implications for clinical practice and patient care.
6.Consider adding a paragraph on the limitations of the study. Mention any potential biases, confounding factors, or limitations in generalizability. This will help provide a balanced view of the research findings.

Additional comments

1.The mention of hip fractures being more prevalent in the elderly, particularly women over 50 years old, adds an intriguing element to your study. To further substantiate this claim, it would be marvelous if you could provide compelling statistics or verifiable references, thereby solidifying the foundation of your research.

Reviewer 2 ·

Basic reporting

Overall, the content of the manuscript is expressed in clear and technically correct English. The introduction and background effectively address the broader field of knowledge by discussing the significance of POD as a cognitive impairment and highlighting the relevance of predicting and managing POD in improving patient outcomes.

Experimental design

The study fills a specific knowledge gap in understanding the factors that contribute to the occurrence of POD in elderly hip fracture patients. By incorporating both age and DARS scores into the multivariable logistic regression analysis, the authors provide valuable insights into the importance of advanced age as a significant risk factor for POD occurrence. Furthermore, the study demonstrates the potential of utilizing the DARS score to predict POD and highlights the limitations of using BIS values alone.

Validity of the findings

See below

Additional comments

1) The explanation of what the AUC (Area Under the Curve) represents and its interpretation in the context of this study is lacking clarity. The meaning of sensitivity and specificity and their relationship to the optimal cutoff value should be described in more detail, providing a clearer understanding for readers.
2) The provided context on the anesthesia depth monitor used to detect MACbar and ETAC is insufficient. More information is needed to explain how the monitor works and establish its reliability in measuring anesthesia depth.
3) The language needs revision by a fluent English speaker, companied with a certificate of language editing service and a manuscript with tracked editing records as a Suppl.
4) State n = ? in Section Statistical analysis and all figure legends.
5)The discussion is not in-depth, limitation is lack, and the clinical implications shall also be added.
6)Must use institution emails for corresponding authors in the title page, as the corresponding author is in a university-affiliated hospital.
7) Please provide a duplicate check report by authors as a supplementary file (total < 20%, each < 2%).

·

Basic reporting

Overall the written English is clear, unambiguous and technically correct for the most part except with the following exceptions:

1) Postoperative Delirium is abbreviated as POD. This is a initially confusing because POD is the typical abbreviation for "Post-operative Day". Especially when writing a study about patients' post-operative condition, this causes confusion and going back to the original study paper by Cooter et al. that this study is modeled on, they too avoid that terminology.

2) Grammatical errors:

In the abstract:
Line 12: "after the hip fracture surgery" should be "after hip fracture surgery"
Line 25: "impacting factors" should be "factors impacting". Suggestion: Multivariate logistic regression analysis found that age and DARS scores were factors impacting post-operative delirium occurrence after hip fracture surgery (P<0.05).
Line 41: Similar to line 12. Should remove “the”

The authors do provide sufficient background context in the introduction however the references used are not direct.
Line 35: The statistic given is for worldwide hospitalization however the reference used (reference 1) is for a national study conducted in China. Since this is a previously published methodology being validated in a Chinese population, quoting Chinese statistics would be appropriate.
Line 43: Authors state that early ‘prediction’ of postoperative delirium after hip fracture surgery has significant importance in improving patient outcome but do not reference if there is any study that shows that prediction of outcome has any significant importance and if there is any intervention that may be implemented.
Line 51: aaMACbar is mentioned. In reference 7 (Cooper et al. ) there is no mention of aaMACbar but just aaMAC. MACbar is MAC that Blunt Autonomic Reflex and that is not what was used in the Cooper et. al study.
Line 103: Reference should be [7] not [9].
Overall there is good flow to the introduction with some room for improvement as above..

Overall, the manuscript structure, presentation of
tables/figures, and raw data sharing meet the standards for a professionally
published article. The data is organized and presented in a clear manner although table 4 may be unnecessary. If these data are to be presented, a single table may be sufficient.

This article presents self-contained results that are relevant to the original hypotheses:
* The introduction clearly states the research aim to investigate the value of DARS for predicting postoperative delirium after hip fracture surgery in the Chinese population.
* The methods describe how all necessary data was collected to address this aim.
* The results section directly addresses the study aim and presents findings relevant to evaluating DARS as a predictive tool.
* DARS scores are compared between POD and non-POD groups.
* ROC analysis evaluates the predictive performance of DARS.
In summary, the manuscript contains all relevant background, methods, results, and discussion needed to evaluate the hypotheses and reach the stated conclusions.

Experimental design

The research question is well defined, relevant, and meaningful, with a clear knowledge gap identified. The manuscript clearly defines the research question, establishes its clinical relevance, explains how it addresses a deficit in the current literature, and conveys the meaningful purpose of the study. The research aims and knowledge contribution are well articulated.

When evaluating the study for investigational rigor, there were a couple of concerns that the authors need to address. It is unclear how patient selection was completed.
- Inclusion criteria stated a population of the age of 60 years but the review of raw data shows 6 patients aged less than 60. Also, the study was meant to be a retrospective study, but one inclusion criteria was that “post-operative follow up could be conducted”. Given the 3 day time frame for diagnosis, it is unclear if this was a prospective study or a retrospective study. It is currently listed as a retrospective study however in order for it to be retrospective, every patient at the study location would have to have had a post-operative follow up, which from the writing does not seem to be the case
- The rationale behind the selection of 151 patients was not stated to determine if the study was adequately powered to find a difference.
- DARS scores were calculated in a standardized, validated way based on anesthesia records.
- Univariate analysis was conducted appropriately however there are some concerns regarding the validity of the multivariate analysis.
Ethical standards:
- The study was approved by the hospital ethics committee.
- Unfortunately given the confusion on whether this study is truly retrospective or prospective, it is hard to determine the conformity with ethical conduct.

Overall, the study methodology, other than the confusion with patient selection and follow up was sound and reproducible.

Validity of the findings

This was a replication study using the methodology previously described by Cooper et. al. and was carried out in a similar manner. I have outlined concerns regarding methodology previously.

The statistical analysis show a P value <0.05 for both age and DARS and multivariate analysis seems like to show that both are independent but I would recommend rechecking. I reviewed the raw data and found a vast difference in the two datasets when it came to age. In order to truly compare the effect of DARS on patients irrespective of age, an age adjusted analysis should be conducted. However given the vast difference in age in both the groups, it is difficult not to count age as a major confounding variable.

Overall given the difference in age in the two dataset, the validity of the results has to be questioned and is a major limitation. The samples are not well matched.

On review of the raw data there were some other interesting observations. Raw data files also show surgical blood loss collected down to the 0.1 ml which seems odd given that collection of such precision in blood loss is difficult. Secondly, patients were limited to a single preoperative comorbidity. No patient had hypertension and diabetes which is also quite odd.

In the discussion portion, there was much irrelevant discussion. Details of how the BIS works are not required. Study quoted in Line 183 talks about how higher BIS levels were correlated to higher incidence of post operative delirium. This study, using DARS is claiming the complete opposite. In line 192-193 it is mentioned that MACbar is used for evaluation due to the significant influence of age on MAC. An explanation and references for this should be provided.

Conclusion states DARS demonstrates good predictive value however, given the age related bias, that conclusion would have to be reworded.

Additional comments

The authors have done good work with this study. Overall, the premise of the study is sound and there is much need to re-validate newer tools in different populations, however there are some flaws in the reported methodology that overshadow the validity of the results. I think clarification and revisions may be necessary before this paper is ready for publication.

---

## Round 0.2 · Minor Revisions

Please carefully read the comments and suggestions from reviewer 3 and provide your responses accordingly.

Reviewer 1 ·

Basic reporting

The authors have addressed all issues. I have no more comments.

Experimental design

The authors have addressed all issues. I have no more comments.

Validity of the findings

The authors have addressed all issues. I have no more comments.

Additional comments

The authors have addressed all issues. I have no more comments.

Reviewer 2 ·

Basic reporting

authors have addressed questions.

Experimental design

authors have addressed questions.

Validity of the findings

authors have addressed questions.

Additional comments

authors have addressed questions.

·

Basic reporting

On my re-review, I found that the authors had made the relevant changes. Literature references were overall improved.

The only I was concerned with was where the authors noted: "aaMAC is calculated based on two indicators, MACbar and ETAC, both of which reflect the effective indicators of inhaled anesthetic depth". I have found no evidence that the above is true. MACbar needs to be defined. The common understanding of MACbar is MAC that Blunts Autonomic Reflexes. Cooter et al. have not used MACbar anywhere in their study and therefore this causes confusion.

Also there were no new tables shown in the revision although the authors have stated that they fixed the issues with the tables.

Experimental design

By changing the inclusion criteria and removing certain aspects of the previous iteration, the study design now makes more sense.

Validity of the findings

Again, the authors fail to recognize the major limitation of their study which is that the most important confounding factor for delirium, age, remained a confounding factor in their study. This MAJOR limitation must be discussed by the authors openly. The authors mention in the discussion "MACbar is used for evaluation due to the significant influence of age on MAC". What is the reference for this?

---

## Round 0.3 · Minor Revisions

Please carefully read the comments from the reviewer. The concerns should be addressed before the paper can be published.

I looked through all the references provided by the Authors. None referred to MACbar except in the following context:

"The concept of MAC-bar is an estimate of the MAC of volatile anesthetic that blocks autonomic responses to surgical incision in 50% of patients. The autonomic responses commonly used to define MAC-bar are changes in pupil dilation, heart rate, and blood pressure.[1] MAC-BAR was determined by measuring the level of catecholamine in venous blood and has been calculated to be roughly 1.5 MAC."

This is not the context that the authors have used it in.

Also note that the authors have referenced "Stat pearls" which is not an appropriate reference.

Age-adjusted MAC is NOT MACbar. Those are two different concepts and I have looked through all the references provided by the authors. I would advise removing the term MACbar from the publication.

·

Basic reporting

I looked through all the references provided by the Authors. None referred to MACbar except in the following context:

"The concept of MAC-bar is an estimate of the MAC of volatile anesthetic that blocks autonomic responses to surgical incision in 50% of patients. The autonomic responses commonly used to define MAC-bar are changes in pupil dilation, heart rate, and blood pressure.[1] MAC-BAR was determined by measuring the level of catecholamine in venous blood and has been calculated to be roughly 1.5 MAC."

This is not the context that the authors have used it in.

Also note that the authors have referenced "Stat pearls" which is not an appropriate reference.

Age-adjusted MAC is NOT MACbar. Those are two different concepts and I have looked through all the references provided by the authors. I would advise removing the term MACbar from the publication.

Experimental design

ok

Validity of the findings

ok with limitation

---

## Round 0.4 · accepted · Accept

The authors have addressed the comments from reviewers.